# The Vulnerability of European Roma to the Socioeconomic Crisis Triggered by the COVID-19 Pandemic

**Almudena Macías León** * and **Natalia Del Pino-Brunet**

Department of Social Psychology, Social Work and Social Anthropology, University of Málaga, 29010 Málaga, Spain
* Correspondence: almudena.macias@uma.es

**Abstract:** The Roma are the most significant ethnic minority in the EU, subject to severe discrimination, social exclusion, and poverty. Due to their deplorable living conditions, isolation, and widespread antigypsyism, Roma are among the most affected by the socioeconomic crisis triggered by the COVID-19 pandemic. This article aims to assess the impact of this crisis on the Roma population from a multidimensional perspective. A thematic review of recent studies and reports on the pandemic's effects on the Roma ethnic minority in Europe was carried out. In this work, the COVID-19 pandemic has been identified as a new global factor that influences the pre-existing exclusion dynamics and Roma mobility within Europe. Results show that these precarious living conditions have deteriorated to alarming levels in most European countries, leading to increased food insecurity and new forms of discrimination and stigmatization. The Roma ethnic minority has been disproportionately affected by mobility restrictions imposed by COVID-19. In all European nations, racist and xenophobic attitudes toward the Roma ethnic minority have increased during the socioeconomic and health crisis. The pandemic has intensified a process of ethnicization, fostering anti-Roma sentiment among the general population.

**Keywords:** (post) COVID-19 pandemic; social impacts; inequalities; ethnicization; antigypsism





## 1. Introduction

Everyone's day-to-day lives have been affected by the coronavirus pandemic worldwide, but those living in poverty and social marginalization face particular difficulties. Every country has experienced a significant economic and social crisis due to the health crisis triggered by COVID-19. However, the impact of this crisis has been unequal depending on the population sector, and the incidence of mortality and morbidity has fallen on specific population sectors exacerbating health, social, and economic pre-existing disparities within societies (Platt and Warwick 2020).

Preliminary studies suggest that some people are more likely to be affected by health, economic, and social crises because of the interaction of certain factors. Membership in an ethnic minority is one of these factors (Platt and Warwick 2020; Matache and Bhabha 2020; Pareek et al. 2020). In England, environmental poverty and ethnicity are linked to a higher COVID-19 mortality rate (Rose et al. 2020). The pandemic's economic effects have affected the population sectors with more precarious jobs (youth, women, and low-paying jobs). Housing situation, employment status, and income level of families have also shaped the impact of the pandemic on the different population sectors (Ocaña et al. 2020).

The COVID-19 pandemic has been identified as a new global factor in this study, contributing to a new scenario influencing the pre-existing exclusion dynamics and Roma mobility within Europe. This work focuses on how the pandemic has affected Roma's precarious housing, health, education, and employment conditions.

The Roma minority is the largest ethnic minority group in the EU. It is difficult to determine the number of people belonging to the Roma ethnic minority in any given

country. It should be noted that data from other official sources are significantly limited for studying the Roma ethnic minority. Its characteristics, as well as the possible administrative obstacles that hinder the registration of this population, make its representativeness in this census lower than the real one (Macias 2022). However, these data allow some estimates of the volume of this population to be made.

The European Commission sets that the European Roma population could be estimated at 10–12 million (European Comission 2020). Most of the European Roma population live in Eastern European countries. Macedonia, Bulgaria, Romania, the Slovak Republic, and the Republic of Macedonia are the countries with the highest percentages of the Roma population (Table 1). The European country with the highest number of Roma is Romania (1.8 million). This figure is much higher than in other European countries. Spain, Hungary, and Bulgaria are other European countries with a sizable Roma population.

**Table 1.** Estimates of the Roma population in the EU.

| Country | Total Population | Average Estimate (CoE Used Figure) | Average Estimate as a % of Total Population |
|---|---|---|---|
| *Romania* | 21,442,012 | 1,850,000 | 8.3% |
| *Bulgaria* | 7,543,325 | 750,000 | 9.94% |
| *Hungary* | 10,008,703 | 750,000 | 7.49% |
| *Spain* | 46,081,574 | 750,000 | 1.63% |
| *Serbia (excl. Kosovo)* | 7,292,574 | 600,000 | 8.23% |
| *Slovak Republic* | 5,433,456 | 490,000 | 9.02% |
| *Czech Republic* | 10,525,090 | 200,000 | 1.90% |
| *"The former Yugoslav Republic of Macedonia"* | 2,060,563 | 197,000 | 9.56% |
| *Greece* | 11,319,048 | 175,000 | 1.55% |
| *Albania* | 3,204,284 | 115,000 | 3.59% |
| *Republic of Moldova* | 3,562,062 | 107,100 | 3.01% |
| *Bosnia and Herzegovina* | 3,760,149 | 58,000 | 1.54% |
| *Austria* | 8,384,745 | 35,000 | 0.42% |
| *Montenegro* | 631,490 | 20,000 | 3.17% |
| *European Union (27)* | | 6,162,100 | 1.18% |

Source: Updated on 2 July 2012. Most estimates include both local Roma + Roma-related groups (Sinti, Travellers, etc.) and Roma migrants. Document prepared by the Support Team of the Special Representative of the Secretary General of the Council of Europe for Roma Issues.

The enlargement process of the EU towards Eastern European countries that took place in 2004 and 2007 increased the Roma population from 1.7 million (EU-15) to 6 million (EU-27) (Laparra and Macias 2009). The EU focused its attention on ethnic minorities in the applicant countries, and the minorities' situation was one of the central negotiation issues in the accession processes of the candidate countries. Progress was made in the Legal Framework and the articulation of specific strategies and measures for promoting living conditions in these countries.

In this context, the "Roma issue" emerged, focusing the attention of the European institutions as one of the main axes in the negotiation process of these aspiring countries. This expression refers to the "process of technical-scientific analysis and social debate on the situation of the Roma and on the political proposals aimed at improving it" (Laparra and Macias 2009).

Within this Strategic Framework, the EU launched the Roma Strategic Framework for Equality, Inclusion, and Participation for 2020–2030 (7 October 2020, E.C.), which addresses Roma inequality within the EU by focusing on equality, inclusion, and participation as horizontal objectives. Education, employment, health, and housing, remain core areas to be targeted by Member States through their National Roma Strategies. Although the Roma population has experienced critical social advances in recent years, their access to social welfare systems in housing, education, services, social benefits, and health remains a priority issue in political and social strategies (Restrepo-Madero et al. 2017).

The Roma ethnic minority in Europe is not homogeneous within the European context, but high levels of exclusion and labor precariousness are present in the Roma community in any country. Therefore, certain common elements seem to shape the exclusion processes of the Roma ethnic minority dealing with structural discrimination and antigypsyism. During the pandemic, this population's previous living conditions deteriorated to alarming levels in most European nations, leading to increased food insecurity and new forms of discrimination and stigmatization (Cârstocea 2022; European Union Agency for Fundamental Rights (FRA) 2020a). Human rights abuses against Roma have occurred once more in the EU, closely linked to the pandemic's ethnicization process (ERRC 2020).

## 2. Methodology

This article aims to assess the impact of the health and socioeconomic crisis triggered by the COVID-19 pandemic on the Roma population from a multidimensional perspective. For this purpose, a thematic review focused on the pandemic's consequences on the Roma ethnic minority within the EU. Data were collected from journals and databases in English and Spanish in the last three years (2020, 2021, and 2022). We searched three electronic databases to retrieve relevant empirical articles: Proquest, Google Scholar, and Scopus.

To identify studies for the review, the following criteria were established:

- Language: Spanish and English.
- Keywords: Roma population, COVID-19, discrimination, and fundamental rights.

Because of the scarcity of sources on these databases, grey literature was also considered a source of information. Therefore, information from European Union Agency for Fundamental Rights (FRA), European Commission, and ERGO Network have been consulted, as well as others from National and International NGOs. The following categories were set to make data interpretation: housing, education, unemployment, health, and discrimination.

## 3. Roma Living Conditions within the EU

The Roma are the most significant, poorest, and fastest-growing ethnic group in the EU. On average, 80% of Roma were at risk of poverty in 2021. Moreover, 40% of households do not have sufficient income to ensure food security for all family unit members (FRA 2022).

### 3.1. Housing: Lack of Equipment, Spatial Segregation, and Overcrowding

Roma face housing discrimination, lack of access to mortgages and loans, high housing costs, overcrowding, lack of improved forms of sanitation, insecurity of tenure, evictions, and territorial concentration/isolation. It is essential to note the segregation experienced by these minorities in cities, which can be a survival strategy that uses communal ties and networks to fill a gap in providing essential services (Molinuevo et al. 2012). Nevertheless, concentrating these minorities in some geographical regions only increases their vulnerability and exclusion.

At least 45% of households lack at least one of the following facilities: toilet, kitchen, shower, and electricity (FRA 2020b). FRA reported that 82% of Roma in EU countries lived in overcrowded households in 2021. In Greece, Hungary, North Macedonia, Romania, and Slovakia, the share of Roma living in overcrowded homes is above 85% (FRA 2022). In the slum settlements, these housing conditions are particularly prevalent.

In the majority of the main areas classified as social exclusion areas in Europe, the majority of the population belongs to the Roma ethnic minority[1]. On the other hand, it should be noted that many house-bound Roma experience acute denial of ethnic identity. Difficulties adapting to housing are often compounded by hostility from their neighbors (Greenfields and Smith 2010). These housing conditions affect the health status of vulnerable groups (Molinuevo et al. 2012).

### 3.2. Education: Institutional Discrimination and Poor Educational Outcomes

Roma students face discrimination and their needs are not met by the schools they attend (Kılıçoğlu and Yılmaz 2018). This exclusion they feel in educational institutions may be one of the reasons that lead them to drop out of school. Roma and Traveler Roma children have the worst educational outcomes of all countries. At least 10% of children do not attend school and less than 10% have completed secondary education. Early school leaving was 67% before the pandemic (FRA 2020b). Furthermore, segregation affects more than half of Roma children aged 6–15 across the EU countries (FRA 2022) and this situation negatively impacts the life chances of children[2].

Roma children faced difficulties accessing technological equipment (such as computers) and the internet as well as significant obstacles in overseeing distance learning (FRA 2020b). Families could not support homeschooling as they lack the digital knowledge required to supervise their children of school age (Magano and Mendes 2021). The exclusion of Roma extends to other spheres of social and political life due to these dynamics in the educational sphere.

### 3.3. Unemployment: High Employment Rates and Informal Activities

High unemployment and informal jobs are characteristic of this population. In countries such as Hungary, Slovakia, or Romania, with a high concentration of Roma, unemployment rates could reach 90% of the Roma population. On average, 43% of Roma aged 20 to 64 were in paid work in 2021. Only 28% of Roma women aged 20 to 64 are employed compared to 58% of Roma men. Moreover, 56% of Roma aged 16–24 were not in education, employment, or training in 2021 (FRA 2022).

Roma families frequently "live from day to day" and survive without protection by engaging in precarious, frequently irregular activities. Main economic activities of the Roma are related to traditional occupations, such as trade, production, and sale of handicrafts and metalworking. Other activities include scrap collection, street selling, and agricultural campaigns. Most of the jobs are informal, short-term activities that belong to the informal economy.

The lockdown has significantly restricted all of these activities and affected the ability of many of these families to earn a living right away. Because these jobs are informal, it has been hard for these people to get help from the formal labor market. Their access to social services has been restricted by formal residence registration formal requirements, particularly in settlements, which has made their situation even worse (Macias 2022).

### 3.4. Health: Barriers to Accessing the Health System and Low Levels of Well-Being

Regarding health, Roma represents one of the most disadvantaged minority groups in Europe and has the worst health outcomes (McFadden et al. 2018). Life expectancy is around 10 years less than the rest of the population. On average, Roma women live 11 years less than women in the general population, and Roma men live 9.1 years less than men in the general population (FRA 2022).

Roma encounter several barriers to accessing the health system, such as bureaucratic processes, discrimination and negative attitudes from health professionals, cultural misunderstanding, language barriers, and low levels of health literacy (McFadden et al. 2018). In addition, Roma are significantly more likely to have a long-term illness, health problem, or disability that limits daily activities or work; it is also reported that this minority group had had more difficulties with self-care, pain or discomfort, and anxiety or depression (Parry et al. 2007 and relatively low levels of well-being Dimitrova et al. 2014).

Living on the margins seriously affects the health of Roma children; 24% of Roma children are malnourished and 25% of children born are of low birth weight. In addition, 20% suffer from bronchitis or pneumonia, 16% suffer from dermatological diseases, 13% suffer from diarrhea or other gastrointestinal disorders, and 5% have underdeveloped motor skills or other disabilities (Loewenberg 2010). Furthermore, ethnic minority groups

show more childhood trauma, specifically physical abuse/neglect and sexual abuse (Berg et al. 2015).

### *3.5. Discrimination and Anti-Gypsyism*

Roma is frequently an outcast group in the societies of the nations where they reside (Kılıçoğlu and Yılmaz 2018), as there is widespread prejudice against this minority group in all of Europe (Romani CRISS 2003). With the 2004 EU enlargement, violent and hate crimes have significantly decreased in Eastern European countries. However, the effects of the crisis—both the refugee crisis of a few years ago and the current crisis brought on by the COVID-19 pandemic—have resulted in the breakdown of social cohesion, an uptick in violent crime, and an increase in hate speech, which has led to new issues for this vulnerable group (European Union Agency for Fundamental Rights (FRA) 2020a).

Structural discrimination is still present, as well as some discriminatory practices and stigmatization processes (Matache 2014). Due to new political formations and social movements opposing further modernization, European integration and immigration promoted the idea of homogenous nation states. They rejected pluralistic democracies, bringing racism, xenophobia, and hatred. However, because Roma have long experienced hardships, such deprivation is frequently perceived as "normality", which furthers their stigmatization and social marginalization. Such social isolation ultimately fuels animosity toward Roma, normalizing their marginalization and escalating anti-Gypsy sentiment.

Therefore in 2005, the EU did incorporate the term "Anti-Gypsyism" into its lexicon. This essentialist ideology gives rise to persistent and violent racism that dehumanizes victims and uses the discourse of hatred and fear (Río-Ruiz and García-Sanz 2020). Anti-Gypsyism has a significant impact on many areas of Roma's life (Marques Gonzalves 2020).

### **4. Vulnerability of Roma during the COVID-19 Pandemic**

The impact of the lockdown of localities and business closures that brought the Alarm State increased the vulnerability and inequality already present in these populations, creating a crisis that "Roma people have ever faced before". To a certain extent, Roma families' ability to survive was immediately disrupted by the lockdown. Mobility restrictions and the suspension of economic activities in different European countries brought about the inability to earn a living for a significant portion of the Roma population. Their primary source of income, which was already a questionable source of income prior to the crisis (European Union Agency for Fundamental Rights (FRA) 2020a), led to an inability to meet the basic needs of many Roma families.

Many Roma were in a social emergency, and obtaining legal assistance from the government was tough. Contrary to the widespread belief that Roma families depend on these benefits, the Roma households in extreme poverty that received social welfare benefits such as minimum income were marginal.

The pandemic and the contingency measures developed by different governments have had a disproportionate effect on the living conditions of the Roma ethnic minority in Europe, increasing the levels of poverty and food insecurity (Gay y Blasco and Camacho 2020). Access to health services has been further restricted. This new factor has intensified specific pre-existing exclusion-promoting dynamics and vulnerability of the Roma population in every European country. The COVID-19 pandemic has heightened the existing discrimination and stigmatization of Roma (Korunovska and Jovanovic 2020; Creţan and Light 2020). Once again, we witness human rights abuses against Roma in the EU context closely linked to the pandemic's ethnicization process (ERRC 2020).

### *4.1. Socioeconomic Crisis and Multidimensional Impact on Roma Living Conditions*

In terms of housing, they could not fit the hygiene and prevention measures (hand washing, social distancing, etc.) because they needed essential equipment, basic services, and space to implement them. Formal residence registration was required to access social

services and emergency social aid, especially in settlements. Most Roma in those settlements did not fulfill these formal requirements.

Concerning education, the lockdown and adoption of online teaching by schools have increased the digital gap and school failure of children from the Roma ethnic minority. Nearly one third of the Roma children (29%) were not able to do their homework, mainly because they lacked adequate equipment (58.8%) and school supplies (48.7%). In addition, even if they had the means, almost half (49%) did not advance because they needed help understanding the subjects and had no support to solve this issue (Fundación Seretariado Gitano (FSG) 2020b, p. 6). The need for more equipment and technological resources (internet, computers, etc.) in the settlements was even more critical and made it difficult for these children to follow the classes.

Roma have been at risk because of their illnesses and living conditions during the pandemic (Porras et al. 2020). Risk factors included their environment's unsanitary conditions. The situation in slum settlements has been highly complicated because of the previous lack of public healthcare and social services. These conditions mentioned above increased those family nuclei's vulnerability and put many Roma families at high risk (European Union Agency for Fundamental Rights (FRA) 2020a). Mental health should also be considered. The family nucleus members' anxiety levels increased due to the high uncertainty levels and inability to meet their basic needs.

Access to clean water and sanitation, a lack of hygiene products, and cuts to and closures of daycare centers and support services all put community health at risk. Family doctors switched to online and telephone consultations when they implemented telematic assistance. Because of this, it took much work for Roma to receive healthcare, especially for those who needed more digital skills or internet access (Varga 2020).

Low levels of education and literacy among Roma people worsen their disparities in health and use of health services, which are already exacerbated by their general disadvantage and discrimination in day-to-day life, such as inadequate housing or difficult access to jobs (Aiello et al. 2022). Other obstacles associated with other disadvantage factors, such as low literacy rates and prejudice experiences, prevent Roma populations throughout Europe from exercising their right to healthcare.

This process of ethnicization has intensified the pre-existing exclusogenic dynamics materializing in the inability to cover basic needs and food insecurity among the European Roma population.

### 4.2. Roma Mobility as a Risk Factor

It is also necessary to approach Roma migrants within the European context. For decades, the Roma population has moved within and outside of countries (Crowe 2003; Matras 2000; Sobotka 2003). Nevertheless, mobility might constitute a risk factor that makes this group more vulnerable during the pandemic period. The interaction between mobility and ethnicity could have increased vulnerability levels of these migrants.

Freedom of movement is one of the fundamental rights of European citizenship, and institutions have promoted policies to achieve more significant democratic standards. As a result, the EU's expansion to Eastern European countries has made it more likely for Roma to migrate, broadening their horizons.

Significant social and political upheavals occurred during the adhesion process towards Eastern European countries. In the 1990s, Romania's transition from a planned economy to a market economy had a significant social effect on the population and the Roma ethnic minority. During this period of modernization and integration into market economies, the Roma were especially at risk in these origin communities, as they were primarily affected by the rise in unemployment and decreased social benefits, which appeared to reverse the slow integration they had been experiencing under the previous stage (Laparra and Macias 2009).

The population devised several strategies to deal with this political and socioeconomic crisis, including emigration abroad. Europe's 1990s international migrations were not a

new phenomenon. However, the post-1989 economic and political chaos in CEE nations, supposedly on the path to democratization, shaped a new scenario (Crowe 2003).

Roma migration from Central and Eastern Europe to Western Europe increased due to a new wave of violence and prejudice in CEE at the beginning of the 1990s. The Roma's structural discrimination and extreme poverty in their home countries drive this migration. These two factors would promote migration flows in a society that appears to be more tolerant (Sobotka 2003).

Romania was the leading nation in these migration flows. The former Yugoslavia, Bulgaria, Poland, the Czech Republic, and Slovakia were other significant sending nations (Matras 2000). Despite the efforts of European institutions and national governments, these nations needed help to ensure their fundamental rights in their home nations (ERRC 2020).

Migratory flows toward Western European countries have been shaped by the absence of these mechanisms to compensate for the sociopolitical crisis and increased discrimination against this minority. This has been one of the recurring factors in the Roma minority's migration within Europe. The lack of confidence in the institutions' capacity to protect this population's rights has been one of the primary factors in the Roma minority's migration to other regions and times (Matras 2000).

It is difficult to approach the volume of this mobile population within Europe. Formal resources are limited for the study of this population. Its characteristics, as well as the possible administrative obstacles that hinder the registration of this population, make its representativeness in the census and other government resources lower than the real one. Therefore, it is only possible to estimate the number of people belonging to the Roma ethnic minority from Eastern Europe in any given country or region.

COVID-19 made up a new global scenario promoting new elements that shaped the migration patterns of this population. Mobility restrictions and the closure of some municipalities with the Alarm State had also particularly influenced Roma migrants, increasing the vulnerability of this population. Improving their living conditions has been the primary motivation for migration. Due to the impossibility of further developing the various economic activities, many migration projects have been halted due to the current health emergency.

Since the middle of March 2020, it was estimated that 500,000 million people from other countries have returned to Romania due to fear of disease or unemployment levels (Crețan and Light 2020). The Roma ethnic minority is a part of this return, though no statistics exist.

The contagion outbreaks in Romania and Slovakia appear to be brought about by the return of thousands of migrants to their home countries. The press and other media amplified the incidents in these Roma communities to make the problem more serious. These discourses encouraged racist and xenophobic behavior in these countries of origin. The pandemic has also ethnicized the population in Romania, escalating hostility toward Roma. On 4 April 2020, the town of Tandarei was quarantined by Romanian authorities due to high infection levels there. With the deployment of police forces in particular neighborhoods and areas, isolated incidents in particular settlements became a matter of social emergency and public health (Crețan and Light 2020).

Similar events occurred in Slovakia, where the government quarantined three locations in the Kosice region (east) with a majority of Roma after dozens of infections were found (ERRC 2020). There have been several instances in which Roma have been accused of bringing the coronavirus to Slovakia.

According to the FRA Report, Bulgarian authorities restricted access to municipalities and neighborhoods populated by Roma. Restrictions based on race, abuse by police, and racist speech have all been condemned in Bulgaria. Even though the authorities rushed to block and quarantine Roma neighborhoods, they did not guarantee that the residents would have full access to clean water, sanitation, medical care, or sufficient food and medical supplies.



Therefore, mobility has constituted a risk factor that makes this group more vulnerable. The pandemic's disproportionate impact on the Roma minority may be influenced by the interaction of these two factors: mobility and ethnicity. Anti-Roma sentiments and discourses have also exacerbated the adverse effects on this population and Roma have become the targets and victims of racist aggression, collective victimization, and ethnic scapegoating.

## 5. Anti-Roma and Discrimination

All European countries have seen increased racist and xenophobic discourses and attitudes toward the Roma ethnic minority due to the pandemic (Korunovska and Jovanovic 2020; European Union Agency for Fundamental Rights (FRA) 2020a; European Comission 2020). Therefore, we can assert that the current pandemic has been ethnicized within the EU, fostering anti-Roma sentiment among the general population. The Roma community is an ideal candidate for channeling fears toward "the others" or "those who are different" in these times of collective hysteria (Berta 2020).

Migrants who have returned to their home countries have been particularly affected by this wave of racist and xenophobic discourse. Fears of the spread of COVID-19 were stoked by the widespread return of migrants in Romania, Bulgaria, Hungary, and Slovakia (European Union Agency for Fundamental Rights (FRA) 2020a, p. 26). This was not a new phenomenon, as Roma have often been claimed to threaten public health (Holt 2020).

"These measures have been fully supported within the legal framework developed to implement the state of Social Emergency, becoming a legitimate practice that hinders the protection of basic human rights", the United Nations Rapporteur states (ERRC 2020).

The ERRC reported violent attacks on Roma communities, including using tear gas against women and children, the disproportionate use of force, the inhumane and degrading treatment of detainees, and police attempts to prevent NGOs from providing humanitarian assistance. Fundamental rights have been violated by the stringent social emergency measures implemented in Eastern European countries (ERRC 2020).

Racist and social hostility toward the Roma ethnic minority has also occurred in Western European countries. In Spain, "Almost 4 out of 10 Roma felt discriminated against or blamed since COVID," according to the FSG study (Fundación Secretariado Gitano (FSG) 2020a). Roma have traditionally been seen as "individuals disintegrated from the Spanish society, unjustly presented as unwilling to adhere to government policies and the mandatory confinement imposed to combat the pandemic". Fake news has also spread through social media in Spain. This could be due, among other things, to the portrayal of the Roma ethnic minority in the media as individuals unwilling to adhere to social norms and, as a result, the confinement guidelines imposed by the State of Alarm. This narrative was based on negative stereotypes dealing with the "rejection of norm-following", the lack of "self-discipline", the "hygienic culture", and the "lack of sense of responsibility" (Berta 2020).

The accusatory discourse regarding Roma's spread of the virus has exacerbated the vulnerability and poverty of Roma communities caused by structural racism. Violence, intimidation, and political and institutional speeches legitimizing discriminatory practices have emerged from anti-Roma rhetoric.

## 6. Discussion

In the European context, discrimination and exclusion dynamics have continued to be significant aspects of the Roma experience. The COVID-19 pandemic has been seen as a new global factor in this study, contributing to a novel scenario and altering pre-existing exclusion-promoting dynamics. This new global pandemic factor and the various emerging elements should be considered in this setting of precariousness and discrimination.

The pandemic and the various government contingency plans have disproportionately impacted the living conditions of the Roma ethnic minority within the EU, increasing former dynamics of exclusion and a process of stigmatization of the Roma ethnic minority in Europe (ERGO Network 2020).

The main economic activities Roma carried out have been significantly restricted by the lockdown, affecting the ability of many families to earn a living. These individuals have found it challenging to claim any assistance or benefits from the formal labor market due to the informal nature of these jobs (European Union Agency for Fundamental Rights (FRA) 2020a).

Furthermore, the accusatory discourse regarding Roma's spread of the virus has exacerbated the vulnerability and poverty of Roma communities caused by structural racism. Violence, intimidation, and political and institutional speeches legitimizing discriminatory practices have resulted from anti-Roma sentiments and prejudices. Right-wing populism and its exclusionary appeals to authoritarianism, ethnonationalism, and cultural grievances related to globalization's mobilization of people, capital, and culture (Inglehart and Norris 2016) are behind those speeches and discourses. Therefore, the Roma ethnic minority has been disproportionately affected by mobility restrictions imposed by COVID-19. Those measures that have affected the entire population have violated the Roma ethnic minority's fundamental rights in the EU context.

In addition, the situation of the Roma migrant within the European context should be particularly considered. The migration patterns of the Roma ethnic minority have changed over time in response to a global order that excludes them and places them on the fringes of the social and economic order. However, in the context of the global pandemic, this population's mobility is an additional risk factor that makes this population more vulnerable. The mobility of this population is not a new phenomenon; nevertheless, the interaction of this factor with those emerging from the global pandemic scenario has increased the vulnerability of this population. According to Bauman, globalization impacts all members of society but not all in the same way. Today's society's social stratification is primarily explained by its members' access to global mobility. The degree of mobility or the freedom of individuals to choose where they want to stay is the factor that differentiates today's consumer society. These always impact the same population segments—those most at risk and with lower social status due to global dynamics (Bauman 2000).

## 7. Conclusions

Despite the European Union's policies and legal framework for advancing the Roma minority's fundamental rights, it does not appear that sufficient mechanisms exist to ensure the rights of these European citizens. European Framework and national strategies have had a limited impact on the situation of the Roma population within the EU context (FRA 2022).

Measures should have a comprehensive perspective addressing exclusion dynamics and structural discrimination. Regarding exclusion dynamics, policies should concentrate on education, housing, health, employment, and social protection.

In the education sphere, stopping Roma from dropping out of school is essential. It is recommended to work toward Roma students' educational normalization, supporting their enrollment, persistence, and future advancement to attain higher education levels. We also see the need to intervene with families, teaching them the importance of encouraging their children to study as a springboard toward inclusion in society and instilling in them the importance of education and patterns to follow to have an adequate follow-up on their children's schooling.

Regarding housing, comprehensive housing and infrastructure programs should become a priority to tackle the overcrowding and poor-quality housing conditions of Roma communities. Social environmental justice and empowerment of Roma could be possible future solutions for solving current inequalities (Málovics et al. 2019).

On the other hand, health promotion measures should be developed to promote healthy habits. It is necessary to establish areas where citizens can quickly access and locate these health resources and bring them closer to these communities.

In employment, recognizing workers who are part of a non-formal economy could be a relevant initiative in a socioeconomic crisis. The government should also support informal



education programs targeting this population to promote their qualification and employment opportunities. Other actions could be oriented to enable the formal participation of Roma in the labor market as a route to economic inclusion and empowerment (Crețan and Powell 2018). National governments should provide more significant support for expanding social protection to Roma, employing advisory services and financial incentives during the post-crisis period if needed.

The recent rise in extreme rights in European and non-European nations must be mentioned if we concentrate on discrimination. Racism and xenophobia have risen due to this ideology, severely affecting ethnic minorities worldwide, including Roma. In this sense, academia and researchers could play a key role in challenging beliefs, attitudes, and harmful narratives before they develop into hatred to generate a healthier community and an atmosphere where hate crimes are less likely to occur. It could aid in creating communities respectful of tolerance and diversity (Collard 2023).

The post-2020 EU Roma Strategy should be scaled up to put combating antigypsyism at its center to encourage all member states to legally recognize this specific form of racism in their domestic law. EU institutions and national governments must officially recognize the existence of antigypsyism in all its manifestations and dimensions, including institutional varieties (Matarazzo and Naydenova 2019). The new EU Roma strategic Framework for Equality, Inclusion, and Participation focused more intensely on combating antigypsyism.

Those holistic measures/programs should be developed and co-ordinated within the European context by fostering transnational co-operation so that this minority can be integrated into society with equal opportunities within the European context.

**Author Contributions:** A.M.L. and N.D.P.-B. contributed to the study conception, design, material preparation and data collection. Data analysis was performed by A.M.L. The first draft of the manuscript was written by A.M.L. and N.D.P.-B. All authors commented on previous versions of the manuscript. All authors have read and agreed to the published version of the manuscript.

**Funding:** This research received no external funding.

**Institutional Review Board Statement:** Not applicable.

**Informed Consent Statement:** Not applicable.

**Conflicts of Interest:** The authors declare no conflict of interest.

## Notes

[1] Gypsy ghettos with no way out. El País. 31 August 2019.

[2] Council of Europe, Commissioner for Human Rights, Fighting school segregation in Europe through inclusive education, position paper, Strasbourg, Council of Europe.

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
