# Peer review of "The Vulnerability of European Roma to the Socioeconomic Crisis Triggered by the COVID-19 Pandemic"

_socsci, doi:10.3390/socsci12050292_

Round 1

Reviewer 1 Report

Social Sciences

The vulnerability of European Roma to the socio-economic crisis triggered by the Covid-19 pandemic.

This is a significant issue which requires critical research. I hope the author will consider the following. I suggest restructuring this work. At the moment it is difficult to follow. An article should tell a good story. Set the scene. At the moment you have a useful table sitting under a heading on living conditions before the pandemic. Think through each of the living conditions you want to discuss. Ensure they have a clear topic sentence, then elaborate and reference. The next section on vulnerability of Roma during the pandemic could then pick up on each of the points from the previous section to elaborate on the effects of the pandemic on exasperating each of these (or not).

I suggest reviewing recent literature also on the topic – a few of these I have found:

Aiello E, Khalfaoui A, Torrens X, Flecha R. Connecting Roma Communities in COVID-19 Times: The First Roma Women Students' Gathering Held Online. Int J Environ Res Public Health. 2022 May 2;19(9):5524. doi: 10.3390/ijerph19095524. PMID: 35564919; PMCID: PMC9102317.

https://www.europenowjournal.org/2021/04/01/covid-19-systemic-racism-and-saying-roma-lives-matter/

https://www.thelancet.com/journals/laninf/article/PIIS1473-3099(20)30381-9/fulltext

Researching the health and social inequalities experienced by European Roma populations: Complicity, oppression and resistance (2021). https://doi.org/10.1111/1467-9566.13411

Collard, M. (2023) Hate in the time of the Covid-19 pandemic: dehumanisation as a side effect; re-humanisation as a remedy. Crime Law Soc Change (2023). https://doi.org/10.1007/s10611-022-10073-8

Edits I would suggest making – whole paper needs to be checked for these.

L. 24 Any country – do you mean ‘every’ country?

L. 29 not sure about this statement – of course there was limited literature on how the pandemic impacted on minority groups in 2020 – the pandemic was just at its initial stages. I would remove this as it doesn’t add to your work. Start instead from Preliminary studies…

L. 38-39. This sentence is not a paragraph and sits on its own. Link it into one of the paragraphs either above or below.

L. 41 Will focus or did it focus – check your tenses throughout.

L. 43 The line that references Bauman does not make sense. Which country did they leave and what global order?

L. 47 This needs a reference – how do you know?

L. 55-63 sentences incomplete, poor grammar, unclear meaning

After reading through this up to L. 64 with the statements about Roma being the largest minority group in Europe – which statistical data are you using? Are you discussing the European Union or Europe.  Is there some debate about the Turks being the largest non-indigenous minority group? Not all European countries are part of the union. Perhaps you need to be clear on the distinction between European countries and those who are part of the union.

L.153-158 another one sentence as one paragraph.

Author Response

Dear Reviewer,

Thank you for your revision and suggestions that have improved my manuscript considerably. I will reply to your revision aspects. 

The work has been restructured. A topic sentence has been incorporated in the sections regarding the living conditions of Roma before the pandemic. We have also tried to make a more cohesive text linking the different arguments from previous sections. 

Recent literature has also been reviewed: 

Aiello E, Khalfaoui A, Torrens X, Flecha R. Connecting Roma Communities in COVID-19 Times: The First Roma Women Students' Gathering Held Online. Int J Environ Res Public Health. 2022 May 2;19(9):5524. doi: 10.3390/ijerph19095524. PMID: 35564919; PMCID: PMC9102317.

https://www.europenowjournal.org/2021/04/01/covid-19-systemic-racism-and-saying-roma-lives-matter/

https://www.thelancet.com/journals/laninf/article/PIIS1473-3099(20)30381-9/fulltext

Researching the health and social inequalities experienced by European Roma populations: Complicity, oppression, and resistance (2021). https://doi.org/10.1111/1467-9566.13411

Collard, M. (2023) Hate in the time of the Covid-19 pandemic: dehumanisation as a side effect; re-humanisation as a remedy. Crime Law Soc Change (2023). https://doi.org/10.1007/s10611-022-10073-8

Regarding edition suggestions, we have checked and changed the following: 

  1. 24 Any country – 'every' country.
  2. 29 Statement has been changed starting from "Preliminary studies…."
  3. 38-39. The sentence has been Linked to one of the paragraphs
  4. 41 Tenses have been checked.
  5. 43 Bauman references have been suppressed. 
  6. 47 Reference incorporated. 
  7. 55-63 Grammar has been improved. 

           - We have tried to clarify the discussion (EU/Europe). 

L.153-158 The sentence has been Linked into one of the paragraphs

Looking forward to hearing from you

Reviewer 2 Report

This study is interesting and should be considered for publication. The paper has a good data interpretation, but there are several sections which have to be improved.

First, the introduction should also present what this paper brings new in existing Roma Studies literature and which particular elements bring new aspects in current Roma literature of Europe..

Second, the literature review needs to be improved with other recent Roma studies in Romania. For instance,  where authors talk about the impact of COVID-19 on the Roma it has to be mentioned that political parties used in pandemic times poor people and (their) sensitive environmental background for populism and electoral gains (see Waldron Richard - doi - 10.1177/0308518X211022363, see Doiciar C. et al, 2021 in Geographica Pannonica, or see Light D, 2020- a study on transnational migrants and Roma outsiders). Also, there is the case of the poor Roma people vulnersbility, discrimination and stigma in Europe - see doi 10.1080/1070289X.2021.1920774 and doi 10.1111/1468-2427.13053 and see studies of Michele Lancione, Ryan Powell and other specialists in Roma Studies. Fleeing from poverty and connected stigma issues against poor migrants were reflected also in other studies - see Wang, 2000 in journal Housing Studies, while for poor people including the Roma see O Brien T. et al, 2022 in journal Identities. 

Method section is too short. It needs to present how were those sources selected and how the interpretation of the data was made. Also, limitations of the data should be presented.

Discussion should be reshaped as there are very few connections between the results of this study and the international literaure.

Conclusions should also include the international implications of this study or how the results of this study bring additional value to what we know in current Roma Studies literature as well as how other studies could further develop the outcomes of this study.

Author Response

Dear Reviewer,

Thank you for your revision and suggestions that have improved my manuscript considerably. I will reply to your revision aspects.

The introduction has been partially modified, incorporating new references. Literature has been reviewed, mainly those works dealing with the impact of COVID-19 on the Roma and the uses of political parties.

The methodology section has been improved, incorporating some eligibility criteria and data interpretation categories.

The discussion has been reshaped, connecting the results of this study and the international literature.

Conclusions have been further developed, incorporating some international policy implications and highlighting “how academia and researchers could play a key role in challenging beliefs, attitudes, and harmful narratives”.

Looking forward to hearing from you

Round 2

Reviewer 1 Report

Second review Social Sciences Vulnerability of European Roma

The methodology states this article aims to assess the impact of the health and socioeconomic crisis trigged by the Covid pandemic on Roma through a thematic review of literature. What is the three-year period of literature drawn on? It is great to see more relevant literature in the reference list – however these are not drawn on in each of the identified themes that are focused on. There needs to be robust referencing of key ideas presented. At times the author links to previous literature – this is fine if the significance is made clear. There are many disjointed short paragraphs that need to be crafted into the work.

p. 2 L. 50  - could reach 10-12 million when?

p. 2. L. 52 – most of who live in Eastern European countries?

p. 2 L. 64 – I don’t know what the enlargement process of the EU means – please give the reader some knowledge.

p.4 L. 121 – you make a shift here to previous conditions and thinking before Covid (outside of the scope of your literature review) – ensure you make the relevance explicit – such as – Significant to this is the work of Molinuevo et al. (2012) who noted the segregation experienced….

p. 4 L. 136. Education: Institutional discrimination – there is no literature here from during the three years of the scoping. What is the situation now? Your argument needs to link what happened previously to the condition/situation now.

p. 4. L. 152. Unemployment. I think you could expand on this – you have some good links to very real data – what were the working conditions due to covid at this time? How did lockdown protocols impact on those in informal labour – such as ‘begging’.

Discussion – you need more work on each of the themes with explicit links to relevant literature before you can enter into a discussion.

Conclusion – there seem to be new ideas and literature being introduced here which should be included much earlier when discussing the themes as this the data of the literature review.

Author Response

Dear reviewer,

Thank you for your suggestions that allow me to improve my manuscript. I will try no reply to these suggestions in the attached document.

Best Regards.

Reviewer 2 Report

Authors have much improved their paper, so I am happy to propose this paper to be accepted for publication.

Just for the production phase of this paper authors have to attentively proofread the paper (for instance, I found in the conclusion section of the paper the expression 'future solucions' but it should be written as 'future solutions') and to attentively cross-check that all the cited references to appear in the reference list of the paper (i.e. I noticed that in the conclusion section it is cited a reference source for year 2018 which does not appear in the paper' reference list).

Author Response

Dear Reviewer,

Once again thanks for your suggestions that allow me to improve my manuscript and shift to the following stage.

I have corrected the expression “future solutions” and have checked the references.

Best.

Round 3

Reviewer 1 Report

This paper is much improved. Thank you for your attention to the suggestions. Please ensure you check through for grammar, spacing, and a final proof read of the whole. 
